# On the Response to Aging of OPEFB/Acrylic Composites: A Fungal Degradation Perspective

**DOI:** 10.3390/polym15030704

**Published:** 2023-01-30

**Authors:** Vladimir Valle, Alex Darío Aguilar, Paola Yánez, Cristina E. Almeida-Naranjo, Francisco Cadena, Jerónimo Kreiker, Belén Raggiotti

**Affiliations:** 1Departamento de Ciencias de Alimentos y Biotecnología, Escuela Politécnica Nacional, Ladrón de Guevara E11-253, Quito 170517, Ecuador; 2Departamento de Ciencias de la Vida y de la Agricultura, Universidad de las Fuerzas Armadas ESPE, Sangolquí 171103, Ecuador; 3Facultad de Ingeniería y Ciencias Aplicadas—Ingeniería en Biotecnología, Universidad de las Américas, Redondel del Ciclista Antigua Vía a Nayón, Quito 170124, Ecuador; 4Centro Experimental de la Vivienda Económica (CEVE)-CONICET, AVE. Igualdad 3585, Córdoba X5003BHG, Argentina; 5Centro de Investigación, Desarrollo y Transferencia de Materiales y Calidad (CINTEMAC), UTN-FRC, Maestro M. López y Cruz Roja Argentina, Córdoba X5003BHG, Argentina

**Keywords:** biodegradation, aging, water immersion, salt fog, natural fiber, polymer composites

## Abstract

Biological agents and their metabolic activity produce significant changes over the microstructure and properties of composites reinforced with natural fibers. In the present investigation, oil palm empty fruit bunch (OPEFB) fiber-reinforced acrylic thermoplastic composites were elaborated at three processing temperatures and subjected to water immersion, Prohesion cycle, and continuous salt-fog aging testing. After exposition, microbiological identification was accomplished in terms of fungal colonization. The characterization was complemented by weight loss, mechanical, infrared, and thermogravimetric analysis, as well as scanning electron microscopy. As a result of aging, fungal colonization was observed exclusively after continuous salt fog treatment, particularly by different species of *Aspergillus* spp. genus. Furthermore, salt spray promoted filamentous fungi growth producing hydrolyzing enzymes capable of degrading the cell walls of OPEFB fibers. In parallel, these fibers swelled due to humidity, which accelerated fungal growth, increased stress, and caused micro-cracks on the surface of composites. This produced the fragility of the composites, increasing Young’s modulus, and decreasing both elongation at break and toughness. The infrared spectra showed changes in the intensity and appearance of bands associated with functional groups. Thermogravimetric results confirmed fungal action as the main cause of the deterioration.

## 1. Introduction

Biodegradation mechanisms of natural fiber-reinforced polymer composites (NFRPCs) are undoubtedly a paramount aspect to be considered in order to further understand the performance of those materials in intended applications [1]. Both polymer matrices and natural fillers are vulnerable to myriad biodegradation processes. As it might be expected, plant fibers, e.g., abaca, coir, flax, hemp, and oil palm, are highly susceptible to be a microbial growth substrate. In particular, bacteria and fungi, whose carbon sources are mainly vegetable materials, are one of the most important chemoorganotrophic organisms that contribute to material biodamage due to their rapid propagation and adaptability [1,2,3].

Microorganisms causing biodeterioration are found in nearly every habitat on earth [4]. Thanks to their adhesion capacity, microorganisms can colonize all types of surfaces, and their growth is affected by several environmental conditions such as temperature, redox potential, pH, and water availability [1,5]. In fact, regarding the latter, microorganisms can be primary, secondary, or tertiary colonizers if the humidity level is low, intermediate, or high, respectively [2,6]. Notwithstanding this, it should be noted that water availability not only is related to the moisture content, but also to the concentration of solutes. In other words, microorganism growth is also influenced by the concentration of salts, carbon dioxide, chloride ions, and sulfates [7,8].

From a broader point of view, the main features of microorganisms that enhance biodeterioration phenomena are intensive metabolite excretion (such as volatile substances, organic acids, enzymes, etc.) and biofilms formation [7,9]. Since biofilms are capable of periodically reestablishing themselves and are not easy to be washed away, they could cause cyclical biodeterioration processes [10,11]. Furthermore, biofilms accumulate water and allow microbial cells to survive adverse conditions [7,12].

Meanwhile, from the perspective of the biodegradation of synthetic polymers, it is not only related to polymer microstructure but also to microbial attack capacity and environmental conditions [1,13,14]. In particular, the biodegradation of acrylic polymers usually presents two stages; both involve reactions catalyzed by enzymes that modify the physico-chemical properties of the polymers [15]. In the first stage, side groups such as alkyls, nitriles, and amides bonding by ester bonds are cleaved from the C−C backbone by hydrolytic enzymatic activities. The enzymes responsible for this modification are amidases, esterases, lipases, and nitrile-degrading enzymes [15,16]. In the second step, the cleavage of the central C−C chain occurs through oxidative mechanisms [15]. Literature surveys focused on the biodeterioration of acrylic polymers have shown that in the case of polyacrylamide, its biodeterioration involves the hydrolysis of amino groups and the cleavage of the central carbon chain. This deamination results in the release of ammonia that is used as a nitrogen source for the growth of microorganisms [17,18].

In the context of NFRPC materials, it has been reported that the susceptibility to biodeterioration of composites elaborated with oil palm empty fruit bunch (OPEFB) fibers increases with the reinforcement content [19]. This behavior is mainly produced by hydrophilic characteristics of OPEFB fibers, which in turn causes more easily adherence of microorganisms to the composite surface up to the point where microbial colonization becomes visible even under harsh atmospheres. Therefore, to some extent, fillers become the main source of food and energy for microorganism growth [19,20].

Several microorganisms are associated with the biodegradation process; nevertheless, fungi are a highly diverse group of heterotrophic eukaryotes living in symbiotic association with natural materials. It is important to take into account that fungal species can also be introduced during processing or even in-service. Furthermore, fungi extract nutrients from cellulose or lignin, which is difficult for other microbes [21,22]. It is well known that these biological agents of deterioration and their metabolic activity produce a number of irreversible changes in materials, namely, porosity increase, loss of cohesive integrity, erosion, etc. [10]. These changes alter the physical, chemical, and mechanical properties of composites [1,23].

In recent years, the attention of the scientific community has been focused on the response of natural fiber composites to performance modification and damage, by means of aging testing such as UV radiation, long-term water absorption, and accelerated weathering tests [24,25,26,27]. In general, these last two are attractive alternatives to emulate extreme service conditions in order to study the biodegradation patterns of composite materials [28]. It is worth noting that salt spray testing is one of the most useful techniques for simulating the natural conditions of coastal environments. Furthermore, it allows researchers to address the influence of saline atmospheres on the properties of composites with greater susceptibility to biodeterioration.

Although numerous studies have reported the use of OPEFB fibers and various polymer matrices in composite development, very few have focused on a facile method to elaborate composites based on this reinforcement and acrylic waterborne polymers, in combination with accelerated aging testing. In this work, OPEFB fiber-reinforced acrylic thermoplastic composites were prepared according to the method proposed in a previous investigation. Since fungi are found intrinsically in natural fibers as well as in both outdoor and indoor environments, the objective of this study was aimed at further studying the fungal biodeterioration behavior of these composites in terms of their weight loss, mechanical, thermal, morphological, and infrared characteristics when subjected to extreme service conditions, simulated by means of immersion in water and salt spray testing.

## 2. Materials and Methods

### 2.1. Materials

OPEFB wastes were collected from the palm oil extraction process of a company located in the Province of Esmeraldas, Ecuador (0°57′33.1″ N 79°39.238′ W). Water-based acrylic thermoplastic resin SINTACRIL A-292^®^ was provided by Poliacrilart in Quito, Ecuador.

### 2.2. Composite Elaboration and Accelerated Aging

In the first stage, OPEFB wastes were dried at room temperature for 24 h and subsequently shredded using a SHINI blade mill, model SG-2348E (Ningbo, China). Thereafter, the fibers were dried at 103 °C for 3 h. In addition, the length of OPEFB fibers was measured using methodology proposed in a previous study [29]. In broad terms, fiber length data were obtained from 40 images through processing with ImageJ^®^ software. The images were acquired keeping focal length (5.58 mm), sensitivity (ISO 100), aperture (f/1.8), and resolution (4608 × 3456 pixels) constant. Fiber length data fitted exponential distribution, as can be seen in Figure 1.

To prepare the composites, OPEFB fibers were placed in an acrylic resin bath with constant stirring (500 rpm) for 30 min. The embedded fibers were dried at 103 °C for 3 h and then processed in a LAB TECH hydraulic press, model LP-S-50 (Mueang Samut Prakan, Thailand), at 150 bar for 40 min. Three processing temperatures were tested: 80 °C, 100 °C, and 120 °C. This method was based on a previous work [30]. The processing temperatures were chosen in reference to the thermal stability of raw materials, and all composite samples were elaborated with the highest content of OPEFB fibers, namely 42 wt.% of filler content and 58 wt.% of resin.

In order to simulate extreme service conditions, three accelerated aging treatments were applied to the composites. In doing so, samples were subjected to the conditions outlined in Table 1. Further details of the water immersion test used in this research work can be found in a previous study. Broadly, the volume of the water bath was controlled and measured at 1 L throughout the test [31]. The saline atmospheres were carried out using a Q-FOG chamber, model CCT600. Figure 2 shows the methodology used in this study.

### 2.3. Fungal Degradation Assessment

#### 2.3.1. Fungal Identification

Microbiological analysis of isolation and identification of fungal strains were carried out for the composites subjected to the three aging tests. For this, samples were taken from four different areas of the composite surface, and each one was placed in 200 mL of sterile distilled water with 0.05 V/V.% of Tween80. Triple dilution banks were made from each sample and planted in Petri dishes with potato dextrose agar (PDA) medium. Afterward, incubation was conducted at 25 °C, and after 3, 7, and 10 days of growth, a different colony of the strains on each plate was selected. Each colony was planted in Petri dishes with PDA medium by the three-point puncture method. The Petri dishes were stored in the incubator at 25 °C to carry out weekly purifications in order to obtain contamination-free fungal strains.

For the identification of the fungal strains, a macroscopic characterization of the fruiting bodies and spores was performed. In this procedure, fungal strains of 14 days of growth at 25 °C were used.

#### 2.3.2. Testing and Characterization

Initially, weight loss determination was performed. For this purpose, samples before and after salt-fog exposure were dried at 60 °C for 48 h and then weighed. Fourier transform infrared spectrophotometry (FTIR) was carried out on the composites in attenuated total reflectance mode, using a JASCO spectrophotometer, model FT/IR-C800 (Tokyo, Japan). The study wavelength range was from 4000 to 400 cm^−1^, with a resolution of 4 cm^−1^ and 20 scans. Thermogravimetric analysis (TGA) was performed, using a SHIMADZU thermobalance, model TGA-50 (Kyoto, Japan), in the range between 20 and 600 °C with a heating rate of 10 °C/min and nitrogen flow of 50 mL/min. In addition, the morphology of the composites was evaluated by scanning electron microscopy under 20 kV, using an ASPEX electron microscope, model PSEM eXpress (Billerica, MA, USA). Finally, tensile behavior of the composites was carried out according to ASTM D 638, in an INSTRON universal testing machine, model 3365 (Norwood, MA, USA). Load cell of 500 N and crosshead speed of 20 mm/min were used as tensile mechanical testing parameters. Tensile samples were obtained from relatively homogenous degradation zones. The mean values of tensile results (tensile modulus, tensile strength, elongation at break, and toughness) were determined using ten test specimens.

## 3. Results and Discussion

After exposure time, results evidenced differences in fungal biodeterioration. Unlike water immersion and Prohesion cycle treatments, continuous salt fog generated fungal colonization on the surface of all composites. As can be seen in Figure 3, no fungal signs were macroscopically identified in samples subjected to both water immersion and Prohesion cycle testing. This finding was corroborated by microbiological analysis of isolation and identification of fungal strains for composites elaborated with the three processing temperatures.

Additionally, it was observed that fungal biodeterioration could not be exclusively associated with the nutrient’s availability in OPEFB fibers, i.e., cellulose and lignin [21,22]. In accordance with the existing literature, fungal growth was also influenced by water, chloride, and sulphate content, as well as by the temperature of aging testing [7,8]. Although exposure time of water immersion was double that of the salt fog treatment, there was no fungal colonization evidence under experimental conditions probed in this work. Furthermore, the results suggested that for water immersion, substrate characteristics were inadequate for fungal growth due to the absence of oxygen, salts, and minerals. On the other hand, the presence of sodium chloride and ammonium sulphate in salt fog tests, at least in principle, generated favorable conditions for fungal growth; however, dry-off at 35 °C for 1 h in exposure cycle seemed to be an important issue affecting fugal spore germination and consequently hyphae formation. Although the Prohesion cycle test included salts, cyclic moisture stopped the rate and magnitude of fungal deterioration.

The surface of composites of Figure 3c showed signs of microbiological degradation (color changes in the material), indicating that a continuous saline environment favored microbial colonization [32]. Due to the fact that the hydrophilic nature of OPEFB fibers increased moisture absorption, microorganisms adhered to the composite surface and used it as a food source for their development [19,33]. Since the scope of this research was the biodegradation understanding of OPEFB fiber-reinforced acrylic thermoplastic composites, the fungal degradation assessment was performed solely for composites subjected to continuous salt fog testing.

### 3.1. Fungal Identification

The presence of microorganisms was verified with the results of the macroscopic–microscopic identification of fungal colonies. Four fungal strains were found on the composites. The macroscopic parameters such as color, texture, and diameter of the colonies were consistent with the development of *Aspergillus* spp. genus. These microorganisms first developed as whitish colonies, then after seven days of incubation, they turned yellowish or brown with a grainy appearance [34]. At the microscopic scale, the spores presented the elementary characteristics of the *Aspergillus* genus; in particular, a non-septate basal foot attached to a long conidiophore with a final vesicle that carries the conidia produced by the phialids [35]. The microbiological identification patterns are shown in Figure 4. Again, since no fungal evidence were observed in samples subjected to both water immersion and Prohesion cycle testing, only the results of samples aged by continuous salt fog are shown.

In broad terms, fungi and fungal spores are the most common microorganisms living in all environments, regardless of location or season [36]. These types of microorganisms are extremely diverse and present higher concentrations in humid places. The presence of water vapor together with warm temperatures of saline environments promote greater proliferation, since they are the preferred environmental conditions for these species [32]. Fungal contamination by *Aspergillus* spp. is a worldwide problem due to the great variety of species in this genus. According to the literature, more than 250 species have been reported, among which those with the potential to degrade organic substrates, particularly of plant origin, stand out. The cell wall of OPEFB fibers is made up of polysaccharides (90%), predominantly cellulose, hemicellulose, and lignin groups. Cellulose is the most abundant compound and interacts with the other groups mainly through hydrogen bonds, which are susceptible to enzymatic degradation [1,37].

Members of the *Aspergillus* genus produce a broad spectrum of enzymes capable of degrading cell walls, and thus the complete degradation of polysaccharides. The main enzymes responsible for the biodegradation of cellulose are endoglucanases, cellobiohydrolases, β-glucosidases, and exoglucanases. Furthermore, both endoglucanases and β-glucosidases are capable of degrading the xyloglucan backbone. All these cellulolytic enzymes have a retention mechanism; in addition, it has been shown that *Aspergilli* can produce them using cellulose, sophorose, cellobiose, glucose, and xylose as carbon sources. However, cellulose—formed by β-D-glucose molecules through β-1,4-O-glycosidic bonds—is the most abundant carbon source, and at the same time, the most susceptible to filamentous fungi of the *Aspergillus* spp. genus, whose enzymes can completely hydrolyze cellulose [38,39]. Similar to the production of glucanases, the literature has reported that when *Aspergilli* use OPEFB fibers as a growth substrate, xylanases and other side-chain hydrolyzing enzymes can also be produced, including arabinofuranosidases, acetyl xylan esterase, and feruloyl esterases [40].

### 3.2. Weight Loss and Mechanical Evaluation

The average mass loss for each formulation is presented on Figure 5. The results suggest that changes were produced by the reduction in OPEFB fiber mat mass. Although acrylic polymer biodegradation is possible, the presence of lignocellulosic reinforcement accelerates biochemical activity of fungi. It has been reported that cellulosic matter is metabolized by fungi, which in turn produce the conversion of natural polymer chains into water, carbon dioxide, and other fluid species [41]. From the results in Figure 5, it is noted that there is no important influence of processing temperature over weight loss. This is to be expected as processing temperature affects mainly matrices.

Figure 6 shows the tensile mechanical behavior of the composites before and after the accelerated aging test. First, it should be noted that composites were elaborated with fibers ranging in length from 200 to 8990 μm which were fitted to exponential distribution (Figure 1). Because of this, we observed an uneven statistic dispersion of tensile properties.

In a broad sense, processing temperature and saline atmosphere influenced the tensile mechanical behavior of the composites. The processing temperature did not produce specific trends in elongation at break and toughness, which was probably due to the lack of uniformity of OPEFB fibers in terms of their diameter and structure (presence of microfibrils and lacunae). Nevertheless, a slight increase in tensile modulus and tensile strength was identified with increasing processing temperature, which suggested gradual evaporation of solvents and water across the composites. Moreover, the increase in processing temperature produced better material compaction and fiber coverage [30].

On the other hand, it was observed that in all composites after the aging test, the elongation at break and toughness decreased, while the Young’s modulus increased. This was because the saline spray caused not only crosslinking but also embrittlement of the polymer matrix, among other modifications [42]. In fact, it can be inferred that the latter influenced more than the former in the case of the downward trend in composite toughness. It is, however, worthy of note that the tensile strength of the composites increased slightly after the aging tests, which could be associated with the heterogenous structure of OPEFB fibers—of non-uniform shape and diameter—that did not allow a homogeneous fiber–matrix interaction throughout the composites [30]. That is to say, the latter induced distortions along the surface of all composite formulations, which reduced the thickness of the exposed samples, triggering the slight increases recorded in tensile strength.

### 3.3. FTIR Analysis

The results of infrared analysis, before aging test, showed several functional groups of the OPEFB fiber and the acrylic matrix. From the information presented in Figure 7a, a broad band was observed in the region between 3500 to 3000 cm^−1^, which was attributed to the stretching of the hydroxyl groups (O–H) of the residual water and the cellulose–hemicellulose of the OPEFB fiber. The band between 3000–2800 cm^−1^ was ascribed to stretching of the C–H group of cellulose and hemicellulose [30]. The tension of the carbonyl group (C=O) at 1730 cm^−1^ appeared as result of d-glucuronic and d-galacturonic acids in hemicellulose. The band observed at 1373 cm^−1^ was related to the C–O stretching of lignin, while the band at 1244 cm^−1^ was associated with the –C–O–C– group of cellulose chains. The bands between 1000 and 950 cm^−1^ were produced due to β-glycosidic bonds between the sugar units in hemicellulose and cellulose [30,43]. Furthermore, characteristic bands of the acrylic matrix were identified at 1730 and 1450 cm^−1^ related to C=O stretching and C−O bending, respectively. Both bands were associated with ester groups of acrylic polymers, while the band of the C−H group at 1145 cm^−1^ corresponded to the bending of the CH_2_ group [30,43,44]. The spectra of the three composite formulations, manufactured at 80, 100, and 120 °C, were mostly uniform in the wavelengths from 4000 to 750 cm^−1^. Nevertheless, the differences between the FTIR spectra were verified by performing a two-sample *t*-test using OriginPro 2019. It was confirmed that there were no significant differences (*p*-value > 0.05) in the spectra of the composites processed at 80 and 100 °C., while significant differences (*p*-value ≤ 0.05) were found between the composites elaborated at 80 and 120 °C, and 100 and 120 °C.

Figure 7b shows FTIR spectra of the composites after the salt spray test. A clear difference was observed in the intensity of the bands at the three processing temperatures before and after the aging test. The two-sample *t*-test indicated significant differences (*p*-value ≤ 0.05) between the spectra of all composite formulations after the salt spray test. The composite elaborated at 120 °C presented fewer changes than the other two composites in the bands around 3500, 2900, and 1200 cm^−1^. This behavior could be due to the fact that at 120 °C, there was a greater degree of compaction of OPEFB fibers with acrylic matrix (better reinforcement-matrix interaction), which reduced the entry of the saline solution into the composite and its effects on the material. Biodeterioration was caused by the increase in humidity and sodium chloride during the salt spray test, since these factors emulated favorable conditions for microbial adhesion and colonization on the composite surface. This caused the decay of the transmittance in the range from 3500 to 3000 cm^−1^ [45]. In this sense, Pană et al. [46] reported that in a glycopolymer, the displacement of the vibration band of the O–H group was related to the attack of microorganisms, particularly as some species of fungi and bacteria were able to modify the molecular arrangement of the polymer matrix through enzyme secretion. This, coupled with the easy adaptability and high colonization rates of microorganisms, could prompt the movement of the molecules inside the polymer matrix over larger distances [46]. Moreover, the absorption of water could be attributed to the hydrophilic nature of the reinforcement because OPEFB fibers have hydroxyl groups forming hydrogen bonds with water molecules. As such, the OPEFB fibers became more flexible due to the plastification effect, which generated changes in the service behavior of the composite material [19,28,33].

Once the microorganisms became attached, biodeterioration began by consuming available carbon sources. It has been reported that the degradation of the hydroxy-propyl acrylate-based glycopolymer begins with the consumption of the sugar oligomer followed by the polymer chain, which was described by the presence of the C=O group at 1730 cm^−1^ (Figure 7b) [46]. The changes in FTIR spectra suggested that the degradation mechanism was initially produced by the destruction of the glycosidic bonds and later by the degradation of the acrylic polymer chain. In addition, this phenomenon was evidenced by the transmittance decrease of the β-glycosidic bonds in the range of 1000 to 950 cm^−1^. This degradation mechanism was consistent with fungal degradation by members of the *Aspergillus* spp. genus, because these species can produce a wide variety of enzymes such as glucanases, which hydrolyze β-1,4-glucosidic bonds in cellulose [38,39,40].

### 3.4. TGA

Figure 8 shows the mass loss of the composites as a function of temperature before and after the aging test. Between 50 and 150 °C, there were minimal changes (0.14–2.42%) in the mass of the composites. This behavior was associated with the evaporation of free and bound water, followed by small variations due to the formation of volatile products. Between 250 and 350 °C, mass losses of 5.37–7.14% and 3.75–4.83% occurred for the composites before and after the aging test, respectively, which were attributed to the degradation of hemicellulose and bio-based acrylic resin backbone [42]. In the case of the tests after aging, there were fewer losses because part of the hemicellulose could be dissolved during the aging process [47]. The next and most significant mass losses, between 350 and 410 °C, reached values of 48.70–49.94% (before aging test) and 46.79–58.84% (after aging test). In this temperature range, it was considered that cellulose and polymer matrix were degraded. The last mass losses, between 410 and 600 °C, were produced by the degradation of lignin. The losses were between 2.90–9.94% and 6.53–10.30%, for the composites before and after the aging test, respectively [42,48].

### 3.5. Electron Microscopy

As can be seen in Figure 9, macro- and micro-cracks were found on the surface of the composites. The sodium chloride caused porosity increment of the composite surface. This phenomenon was linked to the dissociation of NaCl into Na^+^ and Cl^−^ ions, which could spread within the composite structure favoring local damage (especially in the interstitial areas). Due to this, osmotic diffusion of water at the fiber–matrix interface was stimulated, and moisture uptake was further accelerated [28,49]. Moisture significantly weakened adhesion among OPEFB fibers and the matrix because fiber swelling generated internal dimensional changes, which in turn raised various stress fields in the matrix [50]. Furthermore, crack formation could also be associated with a number of different factors such as fiber degradation, hydrolytic reactions, and microbial attack, which further weaken the adhesion between the natural fibers and the polymer matrix. On the other hand, specialized studies have reported that in polylactic acid matrix composites reinforced with flax fibers, natural fibers acted as water transport media. When the fibers were short, they were completely distributed in the matrix and exhibited a high degree of contact with the surface. In such cases, water could more easily penetrate the fiber structure [51].

In studies related to the biodegradation of NFRPCs, several fungi species have been reported in composites subjected to different aging testing. Table 2 presents biodegradation investigations performed with natural reinforcements in different traditional polymer matrices, e.g., polypropylene (PP), high density polyethylene (HDPE), ethylene vinyl acetate (EVA), and ethylene-propylene-diene elastomer (EPDE).

Comparing the results of this work with those of the studies presented in Table 2, we can observe important differences between OPEFB–acrylic composites and the others. From the microbiological perspective, the fungal species identified in the present study differs from the species reported by other authors. Regarding weight losses, the values of OPEFB/acrylic composites were relatively similar to those recorded in wood flour/PP, pine wood flour/PP, and wood flour/HDPE composites. Additionally, some changes were mainly associated with the increase or decrease in tensile modulus, tensile strength, elongation at break, and flexural strength. These differences, at least in principle, appeared linked not only to the nature of the matrix and the reinforcement but also to the aging treatment. It is, however, worthy of note that some fungal degradation characteristics of the studied composites are somewhat similar to other studies. In summary, our results, together with data reported in the literature, suggest the importance of further studies relating to the aging of natural fiber-based composites under aggressive environmental atmospheres, and, more important, to better understand the influence of fungal biodeterioration on the performance of those materials.

## 4. Conclusions

As a result of aging of OPEFB/acrylic composites subjected to water immersion and salt fog environments, macroscopic signs of biodeterioration were observed exclusively after continuous salt fog treatment. As expected, both permanent humidity and salt presence constituted proper conditions for the proliferation of fungal colonies of the *Aspergillus* spp. genus. The filamentous fungi of this genus produced a variety of hydrolyzing enzymes that degraded the cell walls of the OPEFB fibers at different stages. Furthermore, fungal activity was accelerated by the moisture that diffused throughout the composites. Simultaneously, the swelling of the fibers and their subsequent degradation caused dimensional changes in the composites. The presence of these changes generated tension fields that ended in microcracks on the surface of the material. On the other hand, overall FTIR and TGA results confirmed the biodeterioration of the composites produced by fungal action, being lower in the composites manufactured at 120 °C. From the perspective of the tensile mechanical behavior, the biodegradation increased the Young’s modulus (13.0–22.6%) and decreased both the elongation at break (25.2–50.9%) and toughness (17.9–46.1%). These changes were produced mainly by the presence of *Aspergillus* spp. and its hydrolytic action on the glycosidic bonds of the OPEFB components.

## Figures and Tables

**Figure 1 polymers-15-00704-f001:**
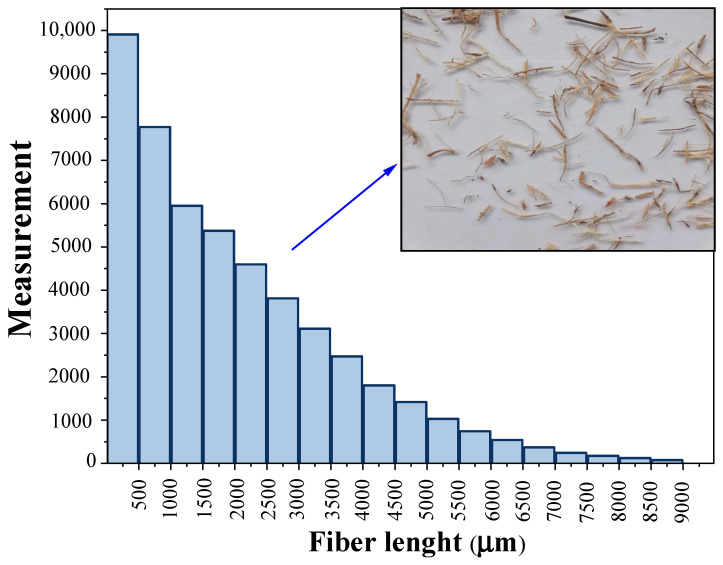
Length distribution of OPEFB fibers used for composite elaboration.

**Figure 2 polymers-15-00704-f002:**
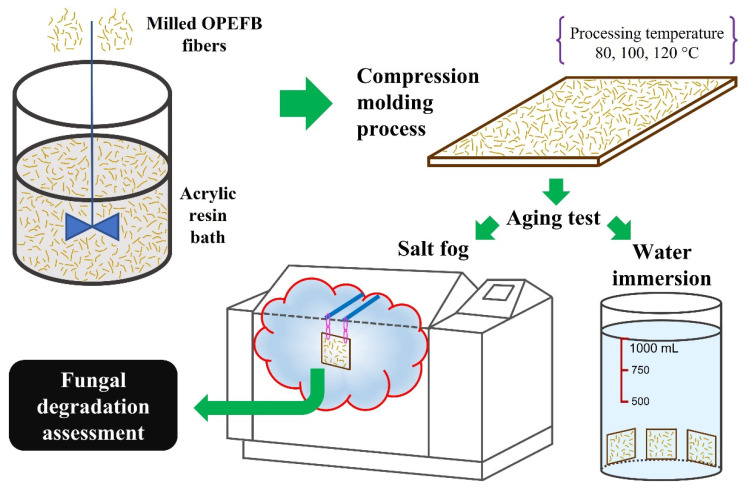
Methodology for the elaboration and accelerated aging of OPEFB/acrylic composites.

**Figure 3 polymers-15-00704-f003:**
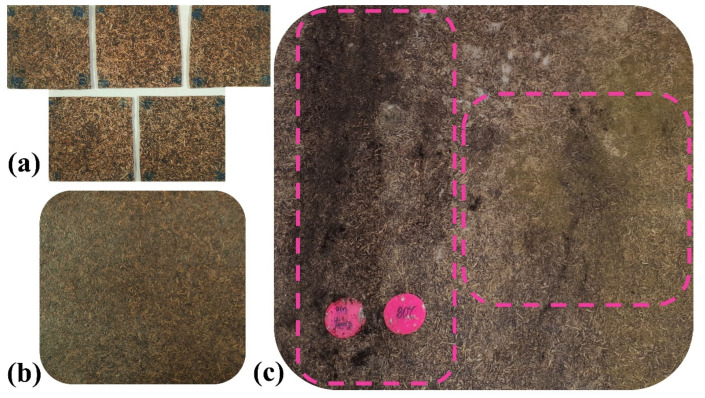
Macroscopic appearance of OPEFB/acrylic composites (Temperature processing = 80 °C) after exposure to (**a**) immersion in distilled water bath, (**b**) Prohesion cycle, and (**c**) continuous salt-fog aging. Dotted lines indicate fungi presence.

**Figure 4 polymers-15-00704-f004:**
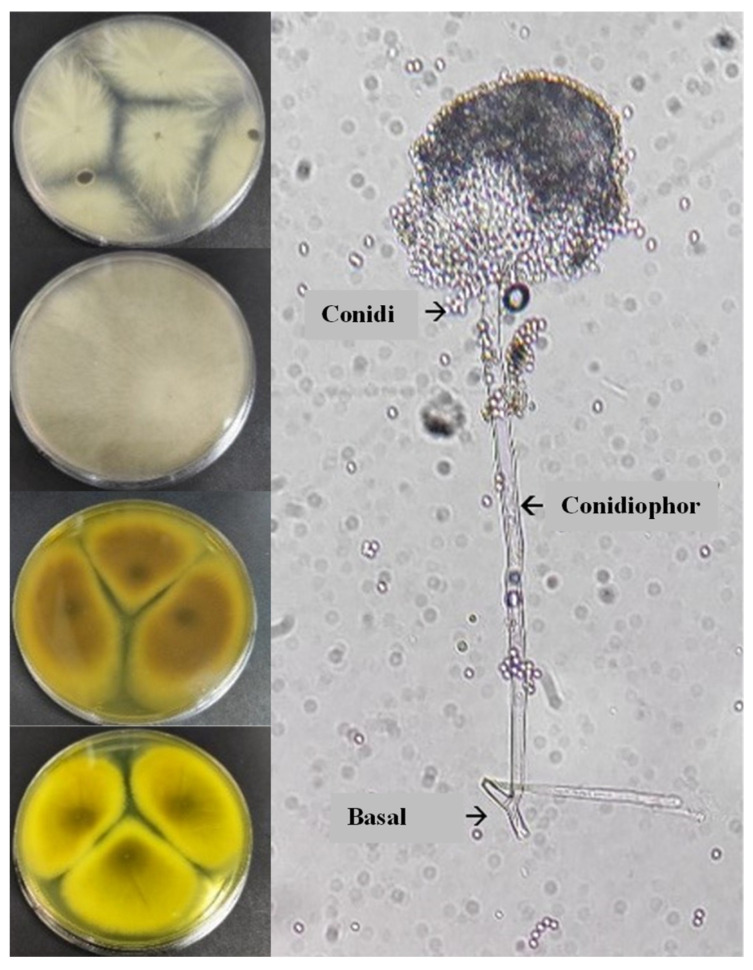
Macroscopic and microscopic identification of isolated colonies on PDA, growth: t = 14 days, T = 25 °C.

**Figure 5 polymers-15-00704-f005:**
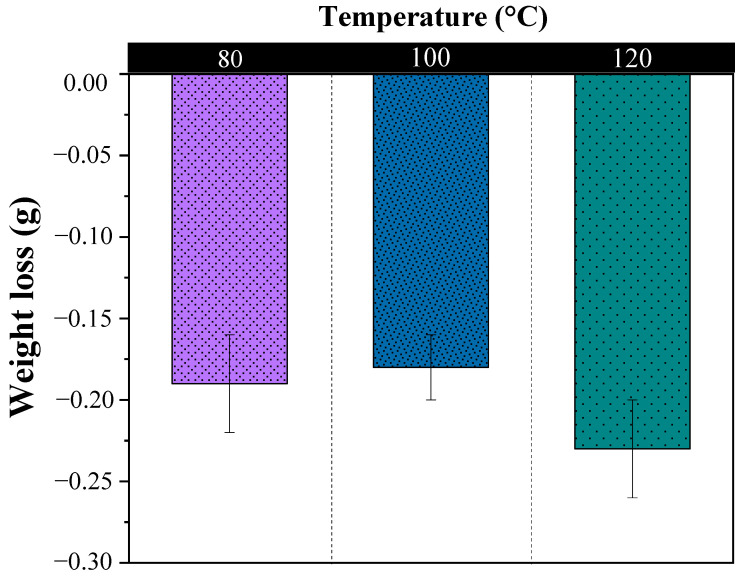
Weight loss for OPEFB/acrylic composites subjected to continuous salt-fog aging. (There were no weight losses in the acrylic matrix without OPEFB fibers as reinforcement).

**Figure 6 polymers-15-00704-f006:**
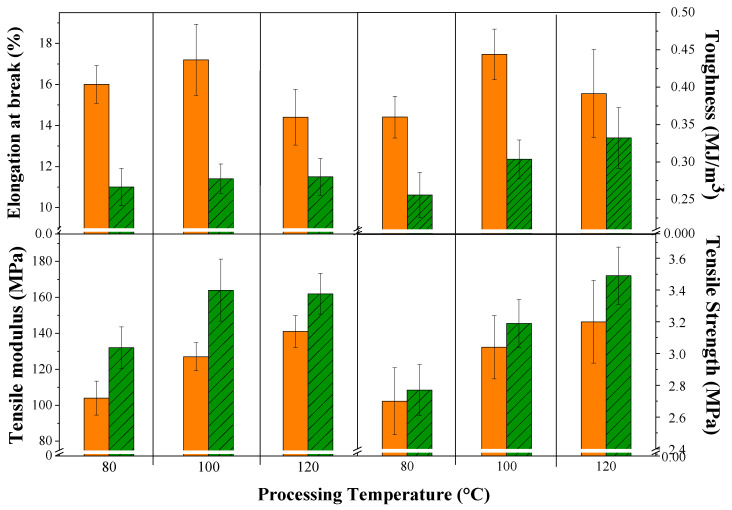
Tensile mechanical properties of OPEFB/acrylic composites. Orange columns (before aging test) and green columns (after aging test).

**Figure 7 polymers-15-00704-f007:**
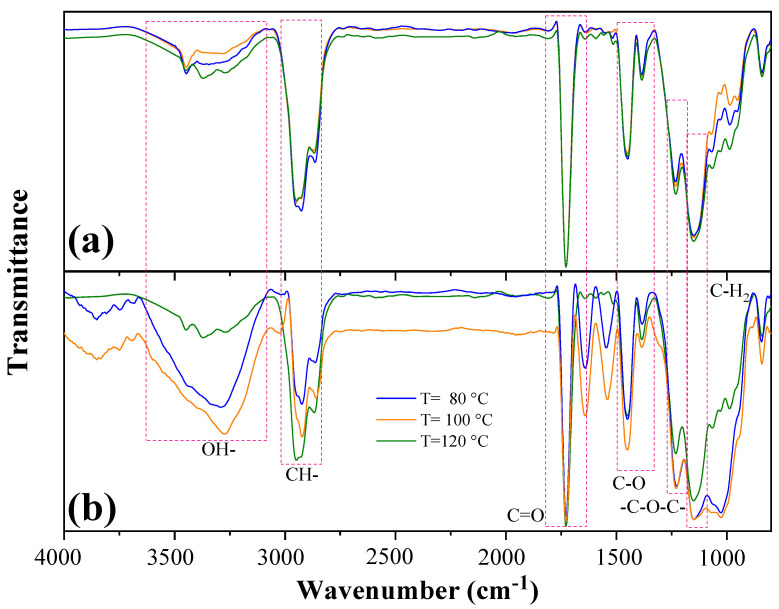
FTIR spectra of OPEFB/acrylic composites (**a**) before aging test, and (**b**) after aging test.

**Figure 8 polymers-15-00704-f008:**
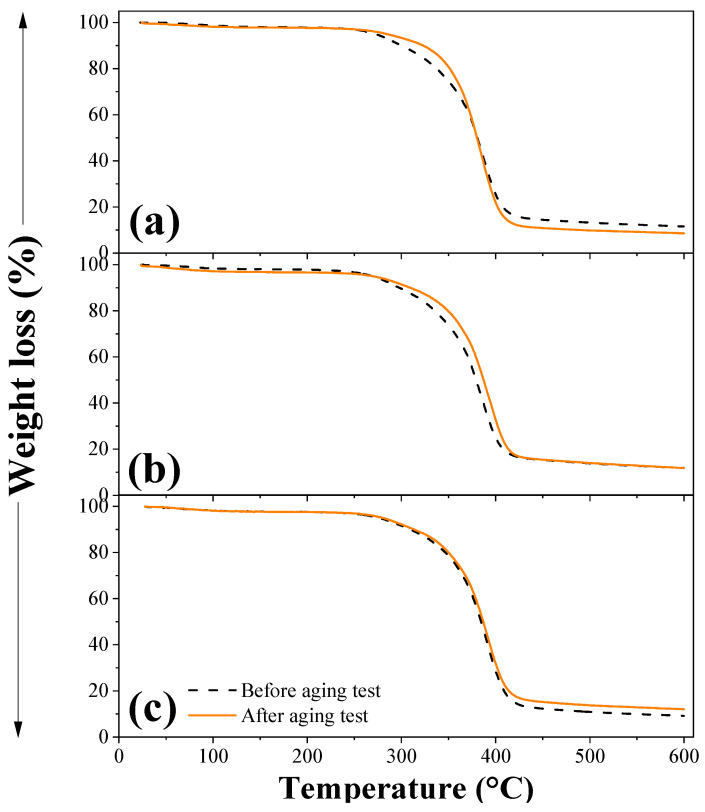
Thermal degradation stages of OPEFB/acrylic composites. Processing temperature (**a**) 80, (**b**) 100, and (**c**) 120 °C.

**Figure 9 polymers-15-00704-f009:**
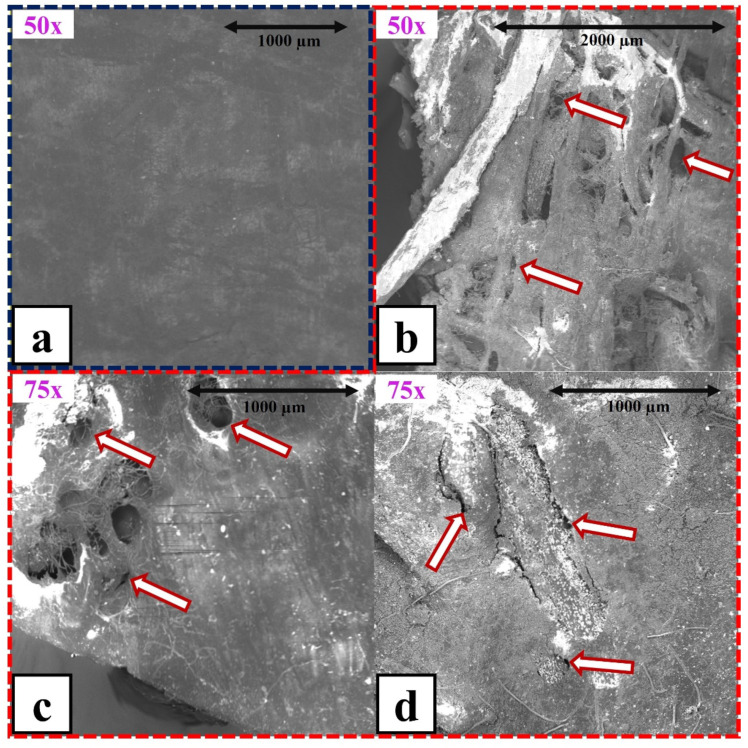
SEM micrographs of OPEFB/acrylic composites: (**a**) before aging test; (**b**–**d**) after aging test processed at 80, 100, and 120 °C, respectively.

**Table 1 polymers-15-00704-t001:** Parameters and conditions of accelerated aging treatments applied to the composites (E.S. = electrolyte solution; E.C. = exposure cycle).

Aging Test	Reference Standards	Conditions *	Sample Dimensions (Length × Width × Thickness)	Exposure Time
Water immersion	ASTM D5529 and UNE-EN-2378	Bath temperature: 23 °C	50 mm × 50 mm × 2 mm	800 h
Salt fog	ASTM G85—Annex 5 Prohesion cycle	E.S.: 0.05 wt.% NaCl + 0.35 wt.% (NH_4_)_2_SO_4_ E.C.: fog (25 °C, 1 h) + dry-off (35 °C, 1 h)	250 mm × 250 mm × 2 mm	400 h
	ASTM B117 **	E.S.: 8.00 wt.% NaCl E.C.: continuous fog (35 °C)		

* Distilled water was used in all aging tests. ** Modified saline atmosphere for greater severity.

**Table 2 polymers-15-00704-t002:** Fungal biodeterioration of other natural fiber-reinforced polymer matrix composites subjected to different aging testing. (W.L. = weight loss; E.B. = elongation at break; T.S. = tensile strength; T.M. = tensile modulus; F.R. = flexion resistance; F.S. = flexural strength).

Composite	Aging Testing	Fungal Identification	Deterioration Effects	Reference
pine wood flour/PP	Modified European standard EN 113	*Coriolus versicolor*	W.L. = 2.5% Increase E.B. = 43.7% Decrease T.S. = 3.1% Increase T.M. = 2.7%	[52]
		*Coniophora puteana*	W.L. = 2.8% Increase E.B. = 8.2% Increase T.S. = 2.4% Increase T.M. = 0.6%	
		*Chaetomium globosumdurante*	W.L. = 0.6% Decrease E.B. = 29.1% Increase T.S. = 0.5% Increase T.M. = 4.6%	
wood flour/HDPE	European standard ENV 12038	*Coniophora puteana*	W.L. = 2.4% Decrease F.R. = 7.6%	[53]
wood flour/PP			W.L. = 2.7% Decrease F.R. = 9.6%	
wood flour/PP	European standard EN 113	*Laetiporus sulfureus + Lenzites betulina*	W.L. = 0.1–1.0%	[24]
hemp/PP	Non-specified	*Chaetomium globosum*	W.L. = 9.8% Decrease T.M. = 38.0%	[41]
flax/PP			W.L. = 10.9% Decrease T.M. = 53.9%	
eucalyptus wood flour/PP-EVA	Method of [54]	*Fuscoporia ferrea*	W.L. = 14.0%	[55]
		*Trametes villosa*	W.L. = 4.2%	
		*Trametes versicolor*	W.L. = 5.4%	
		*Pycnoporus sanguineus*	W.L. = 1.8%	
wood/PP-EPDE	ISO 846	*Aspergillus niger + Penicillium funiculosum + Paecilomyces variotii + Gliocladium virens + Chaetomium globosum*	W.L. < 3.0%	[56]
sugar cane bagasse/HDPE-PP	Agar-plate test method	*Coniophora puteana*	W.L. = 0.2–.6%	[57]
poplar wood fibers/HDPE	AWPA standard E10-06	*Trametes versicolor*	W.L. = −2.0–0.5%	[58]
		*Gloeophyllum trabeum*	W.L. = −0.7–0.1%	
wood/PE	ASTM D2017	*Trametes versicolor*	W.L. = 1.0–6.0% Decrease F.S. = 83.0%	[59]
OPEFB/acrylic	Modified ASTM B117	*Aspergillus spp.*	W.L. = 0.4–2.7% Decrease E.B. = 20.0–34.0% Decrease T.S. = 15.0–32.0% Increase T.M. = 15.0–29.0%	Present work

## Data Availability

Not applicable.

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
