# Peer review of "On the Response to Aging of OPEFB/Acrylic Composites: A Fungal Degradation Perspective"

_polymers, 2023, doi:10.3390/polym15030704_

Round 1

Reviewer 1 Report

3.3. FTIR analysis

Please provide t test for evaluation of differences at specific wavelengths between OPEFB/acrylic composites before and after aging test, T=80oC versus T=100oC, T=80oC versus T=120oC and T=100oC versus T=120oC, because it cannot be concluded whether or not there are differences in the spectra based only on differences (smaller or higher) in intensity.

Reviewer 2 Report

My suggestions are as follows.

1.   The physical structure of the sample after the compression process should be drawn for better understanding.

2.   The information including dimension, thickness, and % weight between OPEFB and acrylic must be mentioned.

3.   The cited methods 29, 30, and 31 were mentioned as “a previous study”. The authors should briefly explain those methods.

4.   What are the novelties of this work compared to references 30 and 31?

5.   In my opinion, the study parameters of this work were very limited. Only 3 processing temperatures were compared, the fiber was not modified or separated, and only the effects of water and salt fog were studied.

6.   Line 255-256, the authors should provide the result of the matrix without OPEFB.

7.   Why was the toughness lowered if the authors claimed that the crosslinking occurred after saline spray?

8.   Line 275-276, the non-uniform shape and diameter of fillers normally give poor mechanical properties of the composites. Why the tensile strength increased in this case?

9.   Figure 5, correct the Y-axis – Tensile strength.

10.                 Line 311-312, the authors should explain the mechanism for the moving of the matrix.

11.                 Line 341-342, what do these values of mass loss indicate?

12.                 Figure 8, the magnification of each micrograph is not the same (50x, 75x)? The scale bar of 8c was not clear and the SEM micrographs for the samples before aging should be added for comparison.

13.                 Line 358-359, the authors claimed that the crack observed in the micrographs indicated fiber degradation and microbial attack. How could the authors be curtained about this? What about poor adhesion between acrylic and fibers?

Round 2

Reviewer 2 Report

-

Author Response

Prof. Dr.

Reviewer of Polymers journal

Quito (Ecuador), December 7th, 2022

Dear Reviewer

We would like to thank you for the review feedback of the manuscript entitled: “On the Response to Aging of OPEFB/Acrylic Composites: A Fungal Degradation Perspective”. We believe that our manuscript has improved substantially and conveys in a better manner some key points of our research.

Best regards,

Vladimir Valle

Departamento de Ciencias de Alimentos y Biotecnología

Escuela Politécnica Nacional

Phone: (+593) 2 2976 300 ext. 2120

Quito, Ecuador.